# Graphene-Based Temperature Sensors–Comparison of the Temperature and Humidity Dependences

**DOI:** 10.3390/nano12091594

**Published:** 2022-05-07

**Authors:** Jiří Štulík, Ondřej Musil, František Josefík, Petr Kadlec

**Affiliations:** 1Department of Materials and Technology, Faculty of Electrical Engineering, University of West Bohemia, 30100 Pilsen, Czech Republic; omusil@students.zcu.cz (O.M.); kadlecp6@fel.zcu.cz (P.K.); 2Centre for Organic Chemistry, 53354 Rybitvi, Czech Republic; josefik@vuos.com

**Keywords:** graphene, temperature sensor, humidity dependence, graphene production methods

## Abstract

Four different graphene-based temperature sensors were prepared, and their temperature and humidity dependences were tested. Sensor active layers prepared from reduced graphene oxide (rGO) and graphene nanoplatelets (Gnp) were deposited on the substrate from a dispersion by air brush spray coating. Another sensor layer was made by graphene growth from a plasma discharge (Gpl). The last graphene layer was prepared by chemical vapor deposition (Gcvd) and then transferred onto the substrate. The structures of rGO, Gnp, and Gpl were studied by scanning electron microscopy. The obtained results confirmed the different structures of these materials. Energy-dispersive X-ray diffraction was used to determine the elemental composition of the materials. Gcvd was characterized by X-ray photoelectron spectroscopy. Elemental analysis showed different oxygen contents in the structures of the materials. Sensors with a small flake structure, i.e., rGO and Gnp, showed the highest change in resistance as a function of temperature. The temperature coefficient of resistance was 5.16^−3^·K^−1^ for Gnp and 4.86^−3^·K^−1^ for rGO. These values exceed that for a standard platinum thermistor. The Gpl and Gcvd sensors showed the least dependence on relative humidity, which is attributable to the number of oxygen groups in their structures.

## 1. Introduction

Since the isolation of graphene in 2004, this material has been extensively studied, and many experiments have been carried out [1]. Graphene has a high potential for usage in many fields. It has been studied in many applications, such as electronics [2], flexible electronics [3], sensors [4,5], biomedicine [6], batteries [7], and water quality monitoring [8,9]. Graphene is a planar two-dimensional material with a hexagonal carbon crystal lattice [1]. Pristine graphene is only one atom thick, but it can be fabricated as a multilayer material [10]. It is considered a revolutionary material that possesses superior properties, such as high surface area, excellent electrical and thermal conductivity, high mechanical strength, and excellent optical properties [11]. Although graphene is a promising material, today, its large-scale use is hindered by difficulties in mass production and quality control [12].

Graphene is produced in many different ways, but two major approaches have been defined: top-down (TD) and bottom-up (BU) methods [13]. TD methods are generally destructive methods where a bulk (carbon) material is broken into smaller sheets of graphene. This approach is considered low cost, but as a result of the breakage of the bulk material, defects are introduced into the final obtained graphene. Typical TD methods are mechanical cleavage, solution exfoliation, ball milling, electrochemical exfoliation, sonication, explosion, and shockwave exfoliation [14]. BU methods, in contrast, are constructive methods. During BU processes, graphene sheets are assembled from carbon atoms. This approach is more expensive and harder to scale up, but the final graphene contains much fewer defects and is close to pristine graphene [14]. The BU methods used to produce graphene are epitaxial growth and chemical vapor deposition (CVD) [15].

One of the many fields where graphene is studied and used in experiments is active sensor materials. The main differences in the relevant research articles are the production/preparation mechanism and graphene active layer of the sensors, which correspond to different material structures. The production aspect is related to BU versus TD methods. The vast majority of authors report that fabricated temperature sensors are characterized by resistance with a negative temperature coefficient (NTC), but some, e.g., Mahmoud et al. [16] and Kim et al. [17], also report a positive temperature coefficient (PTC). Bolotin et al. explained in 2008 that at low temperatures, near-ballistic transport occurs, but when the temperature is increased, the resistivity exhibits two distinct behaviors (NTC and PTC) that depend on the carrier density. When the carrier density is high, graphene exhibits metallic behavior with PTC characteristics. When the density is low, graphene exhibits semiconductor behavior with NTC characteristics [18]. This suggests that the way the sensors are fabricated and modified may result in different behaviors with respect to temperature. However, no comprehensive review or research on graphene temperature sensors of different types (regarding the way they were produced) and their behavior is to be found in the scientific literature.

Temperature sensors are manufactured to meet certain requirements. The main requirements are measurement accuracy, a good response time (e.g., low thermal inertia), and consistent performance under varying environmental conditions (mainly humidity). Because of graphene’s sensitivity toward humidity, it has been utilized by many authors as a humidity sensor, and its parameters have been tested. In 2020, Zhu et al. fabricated a flexible humidity sensor with laser-induced graphene electrodes [19]. Graphene’s advantage of sensitivity toward humidity quickly turns into a disadvantage when other dependences are to be measured. It has been observed that when graphene is used as a temperature sensor, humidity affects the temperature readings [20]. For this reason, it is good to know the humidity dependence of graphene material when we want to use it as a temperature sensor. With a few exceptions [19], this dependence exists in most cases. The reason is that graphene has a number of oxygen groups on its surface (depending on the method of graphene production) that interact with water molecules [21]. A relatively simple solution to reduce the effect of the humidity is the sensor encapsulation. However, this is another additional technological step in sensor fabrication. In addition, encapsulation slows down the heat transfer to the active layer, thus reducing the dynamic properties of the sensor. The graphene humidity dependence has been partially resolved by other approaches. In 2015, Sagade et al. fabricated air-stable graphene-based field-effect devices using Al_2_O_3_; the devices not only showed hysteresis-free behavior and reproducible device characteristics when stored and operated under ambient conditions for weeks but also possessed stable and reproducible doping levels after thermal treatment and exposure to chemicals [22]. In 2018, Romero et al. fabricated an NTC thermistor by using laser pulses for reduced-graphene oxide (rGO) production. The sensor was made stable toward humidity and air by being vacuum sealed between polyethylene terephthalate (PET) films [12]. In 2017, Salvo et al. fabricated a temperature and pH sensor that was unaffected by environmental conditions using a specific type of biomedical-grade polydimethylsiloxane [20]. These examples show that different approaches have been found to solve the above problem.

In this work, four types of graphene-based temperature sensors were prepared. The materials differed in the graphene production method: two of the materials were made by the TD method, and the other two were made by the BU method. In addition to the dependences that characterize temperature sensors, their humidity dependences were measured. This paper aims to compare these four graphene-based materials in terms of their temperature and humidity responses because the structure of the formed graphene affects these properties of the sensor. Moreover, a comparison of several types of graphene in terms of the effect of humidity on the function of the temperature sensor and temperature dependence has not been found in the scientific literature. Based on the scientific literature and the obtained results, we describe and clarify the basic mechanisms related to these sensor dependences.

## 2. Materials and Methods

### 2.1. Sensor Fabrication

The first two materials used as the active layers of the tested sensors, reduced graphene oxide (rGO) supplied by Graphenea (Cambridge, MA, USA) and graphene nanoplatelets (Gnp) from Nanografi (Ankara, Turkey), were prepared from graphene powder. In both cases, graphene powder was deposited as a dispersion, which was prepared by mixing 1 mg of graphene powder with 1 mL of dimethyl acetamide (DMAC). The mixtures were treated for 6 h in an ultrasonic bath to break graphene agglomerates and thus ensure better homogeneity of the dispersion. Next, the dispersions were centrifuged at 10,000 rpm for 45 min to remove agglomerates that had failed to break during sonication. The active sensor layers were deposited from the prepared dispersions using air-brush spray coating. An alumina plate with gold interdigital electrodes (IDEs) was used as a substrate for the temperature sensors. The width of the electrode fingers was the same as the gap between them, namely, 100 μm (see Figure 1). The chemical resistance and stability of alumina and gold are helpful during deposition from organic solvents such as DMAC. The substrate must be heated to 165 °C (boiling point of DMAC). This temperature is necessary for rapid evaporation of small drops of the dispersion during deposition; otherwise, inhomogeneities called “coffee rings” will be formed. These inhomogeneities are formed mainly by slow evaporation of the solvent, during which the solid particles agglomerate again.

Two other types of graphene active sensor layers were prepared by growing graphene on the substrate. Graphene grown from a plasma (Gpl) was prepared by the plasma discharge method, which uses ethanol as the source of carbon atoms and an argon atmosphere, according to the literature [23]. The last type of graphene layer, denoted Gcvd, was prepared by CVD, where polycrystalline copper foil was used as a substrate for the growth of a continuous graphene layer. The copper treatment consisted of annealing at 1000 °C in a hydrogen atmosphere (50 sccm flow) for 20 min inside the CVD reactor. After that, methane was used as a precursor from which graphene grew for 35 min at a flow rate of 1 sscm, and then annealing was performed for another five minutes in a H_2_ atmosphere [24]. Finally, the graphene layer was transferred to the IDE substrate using the nitrocellulose-based technique [25].

### 2.2. Characterization of Graphene Sensors

The graphene layers based on rGO, Gnp, and Gpl were analyzed by scanning electron microscopy (SEM) to verify their structure. A Phenom ProX G6 (Thermo Scientific, Waltham, MA, USA) SEM instrument was used, and images of the layers were created using a backscattered electron detector with an electron beam accelerating voltage of 15 kV at a magnification of 10,000. Then, in conjunction with SEM, energy-dispersive X-ray diffraction spectroscopy (EDS) was used for elemental analysis to determine the amount of oxygen in the graphene structure. A Phoibos 150 (SPECS, Berlin, Germany) spectrometer was used to collect XPS (X-ray Photoelectron Spectroscopy) spectra of Gcvd to obtain information about the amount of oxygen in the structure. The XPS measurements were performed at base pressures lower than 10^–9^ mbar. A monochromatic Al Kα radiation source with a spot size of 1 mm^2^ was used to excite the electrons. The pass energy of the electron energy analyzer was set at 50 eV. The take-off angle between the analyzed photoelectrons and the substrate surface was 90°. EDS could not be used to determine the oxygen content of Gcvd, because this material is only one atomic layer in thickness, precluding the effective use of EDS.

### 2.3. Temperature and Humidity Measurement

All temperature tests were performed in a small in-house-developed measuring chamber. The temperature was controlled by the two Peltier elements in a cascade, which allowed the temperature to be adjusted from −10 °C to 120 °C. During the measurement, the sensors were tightly connected over the entire surface of the sensor substrate to the Peltier cell to obtain the fastest possible response to temperature changes. The humidity inside the chamber during measurements was kept at 40 ± 1% relative humidity (RH) to simulate a standard operating environment. The temperature tests for all sensors were carried out from 0 °C to 100 °C with a 10 °C step change, and each step lasted 10 min. A climatic chamber (VCV3 7060-5, Vötsch, Balingen, Germany) was used to determine the humidity dependence of the prepared temperature sensors. The RH inside the chamber gradually increased from 20% to 90% with a rate of 0.5% per minute, and the temperature was maintained at 25 °C during the whole test. The humidity test was performed at two constant temperatures, 20 °C and 50 °C, to observe how humidity affects the properties of graphene sensors at different temperatures. The electrical resistance was continuously measured during all tests by an E4980A LCR meter (Agilent, Santa Clara, CA, USA). The sensors were contacted via spring test probes connected to the LRC meter using a 4-wire method. The AC signal voltage was 0.25 Vrms at a frequency of 1000 Hz. The resulting changes in the resistance of the sensors during testing with varying temperature and humidity were converted to a relative change in resistance so that the individual sensors could be compared with each other. The formula for the relative change in resistance is as follows:(1)$$Response=\frac{\Delta R}{R_{0}}=\left(\frac{R-R_{0}}{R_{0}}\right)\cdot 100\%$$
where *R* (Ω) is the sensor resistance at the maximum value of the parameter under investigation (T = 100 °C for the temperature test; RH = 90% for the humidity test), and *R*_0_ (Ω) is the sensor resistance at the minimum value of the parameter under investigation (T = 0 °C for the temperature test; RH = 20% for the humidity test).

## 3. Results and Discussion

### 3.1. Material Properties

Figure 2 shows the structure and EDS results of the rGO, Gnp, and Gpl materials. The SEM images of rGO and Gnp confirm that the materials are in the form of small particles, which is crucial for properties such as the type of temperature dependence and conductivity of the active layer. On the other hand, Gpl resembles a diverse continuous network of graphene. The chemical element proportions were determined as the atomic percentage for each element, and the complete numerical results are shown in Table 1. Elemental analysis of rGO reveals, in addition to carbon (47.58%) and oxygen (42.8%), small amounts of aluminum and manganese. These elements in the structure of graphene are most likely residues after chemical reduction reactions of graphene oxide during conversion to rGO [26]. rGO has the largest number of oxygen atoms. The material with the second largest amount of oxygen in its structure is Gnp, with a content of 38.36%, and the rest is carbon, with a content of 61.64%. The material with the lowest oxygen content in its structure, which was determined by the EDS method, is Gpl. The amount of oxygen is 23.97%, and that of carbon is 76.03%.

Gcvd was characterized after formation on a copper foil before the transfer process. Most of the peaks in the binding energy spectrum belong to copper (see Figure 2d). From the peaks of Cu 2p_3/2_ and Cu 2p_1/2_ at binding energies of approximately 933 eV and 952.5 eV, respectively, we can determine the surface atomic content of the material. At approximately 284.5 eV, there is a typical C 1s peak. The typical O 1s peak is very small in the energy spectrum and can be found at approximately 531.5 eV. The surface atomic contents of these three elements were determined assuming a homogeneous distribution of atoms and a Scofield photoionization cross-section. The resulting percentage contents of these elements on the surface are as follows: Cu 2p = 58.96%, which is attributed to the substrate; C 1s = 40.39%, which are the carbon atoms that make up graphene; and O 1s = 0.35%, the oxygen atoms in the graphene structure forming defects. After elimination of the copper substrate, the percentage of carbon in the structure of Gcvd is 99.14%, and that of oxygen is 0.86%. The amount of oxygen in the Gcvd structure is negligible in comparison with the amount of oxygen atoms in the other graphene-based materials.

### 3.2. Temperature Dependences of Graphene Sensors

The sensor measurements were first performed under different constant DC bias voltages (0 V, 1 V, 2 V) to verify the effect of the bias voltage on the sensor response. Figure 3 shows the response of the Gcvd sensor at different bias voltage levels. It can be seen that the resistance value of the sensor decreases slightly at different bias voltages, but the sensor response to the measured temperature profile remains the same at all bias voltage levels.

The results of the temperature test are shown in Figure 4 and summarized in Table 2, which compares the temperature and humidity properties among the individual graphene-based materials. The response of the sensors to a gradual step increase in temperature from 0 °C to 100 °C is shown in Figure 4a–d. The sensor resistance dependence on temperature is shown in Figure 4e–h. The resistance value at a given temperature is determined as the average of the last 5 measurements at that particular temperature during the gradually increasing temperature test. rGO, Gnp, and Gpl show decreasing resistance with increasing temperature. The structure of all of these materials consists of disordered graphene particles forming the entire sensor’s active layer. In the case of rGO and Gnp, the particles are small graphene flakes. In contrast, the resistance of Gcvd has the opposite temperature dependence as the rest of the tested materials (the resistance increases with increasing temperature). The reason is that the active layer is a uniform layer of graphene. Figure 4h also shows that the resistance of the sensor graphene layer is much lower for Gcvd than for the other materials even though the same substrate with identical electrode spacings was used. rGO and Gnp show a higher response than the other two materials over the entire temperature range, with values of 45.1% and 52.0%, respectively. The response of Gcvd is 27.0%, and the lowest response to temperature change is only 20.5%, obtained with the Gpl sensor. Gpl also has the least stable temperature response, which may be caused by the preparation method. Graphene growing on a substrate in a plasma discharge may have a random arrangement, indeterminate number of defects in the structure, and poor adhesion to the substrate.

While Gcvd behaves like a metal, the other materials investigated in this study exhibit behavior typical of semiconductor materials. This is mainly due to the material structure, which is determined by the graphene preparation method. Pristine graphene in one or a few atomic layers with a minimal number of defects has a zero bandgap. This is reflected in the fact that in certain cases, pristine graphene behaves as a semimetal. In our case, semimetal-like behavior was observed for Gcvd, where the electron–phonon scattering effect dominates in the temperature dependence of resistance [27]. As the temperature increases, the probability of scattering of the charge carrier also increases. This causes a decrease in the mobility of the charge carrier and thus an increase in electrical resistance [28]. The other tested materials show semiconducting properties, where other possible types of scattering, such as scattering on defects, impurities, flake edges, and multilayer structures, can occur and dominate over electron–phonon scattering. This leads to a change from zero-bandgap graphene to a finite-gap semiconductor-like graphene [29]. The temperature dependence of the resistance of these materials is determined by thermally activated charge carriers. As the temperature increases, the probability of transfer of charge carriers from the valence band to the conduction band increases, and the resistance decreases [28]. Figure 4e–h suggest that the dependence of resistance on temperature is nonlinear. With regard to the literature and the measured temperature range (which is above 250 K), Arrhenius-like behavior is assumed and can be expressed by the following equation [30]:(2)$$R=R_{0\;}exp\left(-\frac{E_{a}}{k_{b}T}\right)$$
where *R* is the resistance at temperature *T*, *R*_0_ is the resistance at infinite temperature, *E_a_* is the activation energy, and *k_b_* is the Boltzmann constant. For this type of dependence, a linear relationship between ln*(R)* and 1*/T* is assumed. This linearity is confirmed by the dependences presented in Figure 5. The coefficient of determination (R^2^) is higher than 0.99 for all sensors, which shows good linear dependence. The activation energy was calculated for all sensors, and the obtained values were 40.1, 63.9, 116.5, and 124 meV for Gpl, Gcvd, rGO, and Gnp, respectively. These values are comparable to the activation energies of this type of material reported in the literature [27,30]. The activation energy is closely related to the position of the Fermi level in the band gap of the material. The main charge carriers in graphene are holes (P-type conductivity); thus, the distance of the Fermi level from the valence band decreases with increasing activation energy. Therefore, less energy is required to release the major charge carriers (holes). The values of the activation energies correspond to the magnitude of the sensor response: the larger the activation energy, the larger the sensor response.

The response time of a sensor is defined as the time it takes to reach 90% of the maximum change in resistance for the variable change between two stable levels (in this case, temperature). This time is referred to as *t_90_*. The evaluated response time is defined for the temperature change from 20 °C to 30 °C that occurred in the 20th minute of the measurement. A detail of the temperature change area for all tested sensors and the *t_90_* value labeling is shown in Figure 6. The decreasing sensor responses were converted to an absolute value to compare the sensors together. The Gcvd sensor had the shortest response time of 68 s, followed by the Gnp sensor with 89 s. The rGO and Gpl sensors had the longest response times of 121 s and 125 s, respectively. It should be noted that the *t_90_* of the temperature change (from 20 °C to 30 °C) in the measuring chamber was 59 s.

The characteristic parameter of temperature sensors is the temperature coefficient of resistance (*TCR*), which is described by the following equation:(3)$$TCR=\frac{1}{R_{0}}\cdot \frac{R-R_{0}}{T-T_{0}}\;$$
where *R*_0_ is the initial resistance at initial temperature *T*_0_ (10 °C) and *R* is the resistance at temperature *T* (100 °C). The calculated *TCR* values for all types of temperature sensors are shown in Table 2. The Gnp sensor has the highest *TCR* value, namely, 5.16^−3^·K^−1^, and the Gpl sensor has the lowest value, namely, 1.68^−3^·K^−1^. For comparison, a typical platinum temperature sensor has a *TCR* of 3.92^−3^·K^−1^. The *TCR* parameter indicates the temperature sensitivity of a sensor. The sensitivity of the tested graphene sensors is comparable to that of a conventional platinum sensor; notably, the sensitivities of two of the tested sensors exceed that of the conventional sensor. Furthermore, Table 3 compares other graphene-based materials in terms of *TCR*.

### 3.3. Humidity Dependences of Graphene Sensors

Figure 7a shows the response of the graphene-based sensors to gradually increasing RH from 20% RH to 90% RH at 20 °C. The best results were obtained for Gpl and Gcvd, where the relative change in resistance was approximately 1% over the entire measured range of relative humidity. A higher change occurs in rGO, whose resistance changes by 4%. The largest relative change in resistance (7%) is seen in Gnp. It is worth noting that for Gpl and Gcvd, the change in resistance has a decreasing trend, and for rGO and Gpl, it has an increasing trend. The results for the humidity test at 50 °C (Figure 7b) are similar to those for the test at 20 °C, with several differences. The relative change in resistance is higher for all materials at 50 °C. The Gnp response reaches almost the same value as that of the rGO sensor, with values of 9.3% and 10.1%, respectively. The trend for Gpl changes direction at higher temperatures, and the resistance increases with increasing humidity, with a change up to 1.8% at the maximum humidity value. A more significant increase can be seen only at RH values above 50%. The response of Gcvd first decreases, and at 70% RH, it starts to increase; the total change is approximately 2%. These results correspond to the amount of oxygen in the graphene structure. With a higher content of oxygen groups, the sensitivity of the sensor to humidity is greater.

Reduced graphene oxide and graphene nanoplatelets contain many oxygen atoms in their structure that form hydroxyl, epoxide, and carboxyl side groups on the graphene surface. All of these groups are active sites for binding water molecules. Graphene is a p-type semiconductor; therefore, holes are the main charge carriers. Water is a donor. During the physisorption process of water molecules on oxygen side groups, an electron is transferred to the graphene structure. This causes a decrease in the concentration of holes in the graphene and a related increase in the resistance of the entire active layer. As mentioned above, the active layers of rGO and Gnp were prepared in powder form. This means that the total conductivity of the active layer is determined by, among other conductivity mechanisms, hopping and tunneling of charge carriers between individual graphene flakes. The second important mechanism, which especially occurs at higher RH values, is the intercalation of water molecules between the individual flakes of graphene materials. This causes a further increase in resistance because the intercalated water molecules increase the distance between individual flakes and thus reduce the probability of charge transfer by hopping and tunneling [38]. This effect also caused the greater increase in resistance during the humidity test at 50 °C. Water molecules have more energy at higher temperatures, which causes a greater oscillation of the molecules around their equilibrium position; due to this oscillation, the distance between graphene flakes at higher temperatures could be greater than that at 20 °C.

Despite the fact that Gpl contains oxygen atoms (24%) in its graphene structure, the resistance of the active layer decreases with increasing humidity. This effect is associated with the continuous structure of plasma-prepared graphene and the different charge transfer mechanism in this case. At low RH, water molecules bind to oxygen side groups by physisorption. These molecules are usually bound by a double hydrogen bond, which can occur between hydrogen-terminated groups and oxygen in H_2_O or oxygen-terminated groups and hydrogen in H_2_O. Water molecules bound in this way cannot move freely and form a first physisorption layer on the graphene surface. With increasing humidity, the number of water molecules increases, forming a second and further physisorbed layers. Unlike those in the first layer, H_2_O molecules in the second and higher layers are bound by only a single hydrogen bond; therefore, H_2_O molecules in higher layers can be ionized by an electric field to form H_3_O^+^ (hydronium ion), which serves as a charge carrier. The motion of the charge between water molecules is mediated by the Grotthuss mechanism of proton jumps between adjacent water molecules [39]. The result of this process is a reduction in the resistance of the Gpl layer as the humidity increases. The same mechanism can be used to describe the response of Gcvd to humidity, with the difference that the first physisorbed layer of water molecules is formed at edge defects and domain boundaries in the graphene structure, together forming a so-called percolation network [40].

## 4. Conclusions

Four different graphene-based temperature sensors were successfully prepared, and dissimilarities in their structures were confirmed by SEM characterization, except in the case of Gcvd. Based on the EDS and XPS results, the contents of oxygen and other substances in structures of the tested materials were determined. The temperature dependences of sensor resistance were analyzed by the Arrhenius model, where all sensors showed a linear relationship between ln(R) and 1/*T*. The material with the highest temperature sensitivity and thus the highest *TCR* was Gnp. However, it also had the highest response to humidity at 20 °C, which is due to the number of oxygen groups in its structure. In general, graphene with a flake structure showed a better temperature response. In contrast, the material with the fewest oxygen groups (Gcvd) had the least sensitivity to humidity. This material also exhibited semimetallic behavior due to its uniform continuous structure, which is the reason for its inverse dependence of resistance on temperature compared to those of the other materials. Graphene is a suitable material for temperature sensors for new applications in electronics, such as flexible, stretchable, and wearable electronics, where, among other things, the mechanical properties of graphene are important.

## Figures and Tables

**Figure 1 nanomaterials-12-01594-f001:**
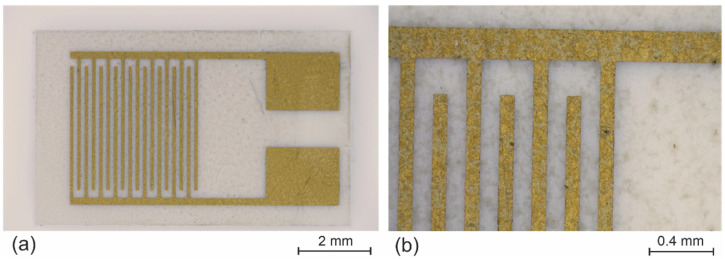
Images of alumina substrate with interdigital electrodes: (**a**) whole substrate; (**b**) detail of the interdigital electrodes.

**Figure 2 nanomaterials-12-01594-f002:**
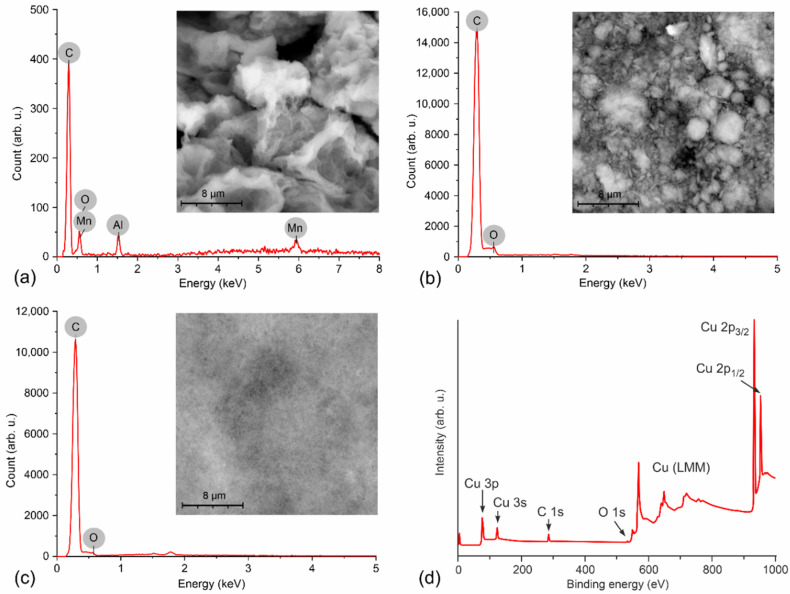
SEM images of prepared graphene layers of the tested sensors and results of chemical element content analysis by EDS for (**a**) rGO, (**b**) Gnp, and (**c**) Gpl and (**d**) XPS spectrum of prepared graphene on copper foil (Gcvd).

**Figure 3 nanomaterials-12-01594-f003:**
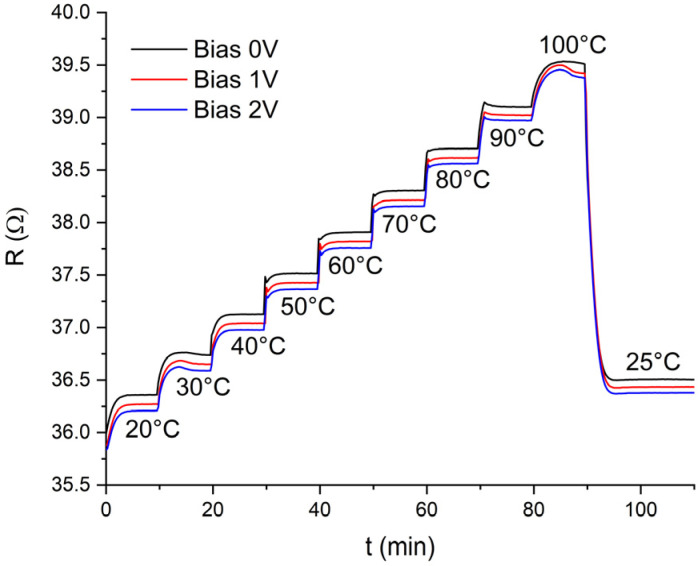
Response of the Gcvd sensor to temperature changes at different DC bias voltage levels.

**Figure 4 nanomaterials-12-01594-f004:**
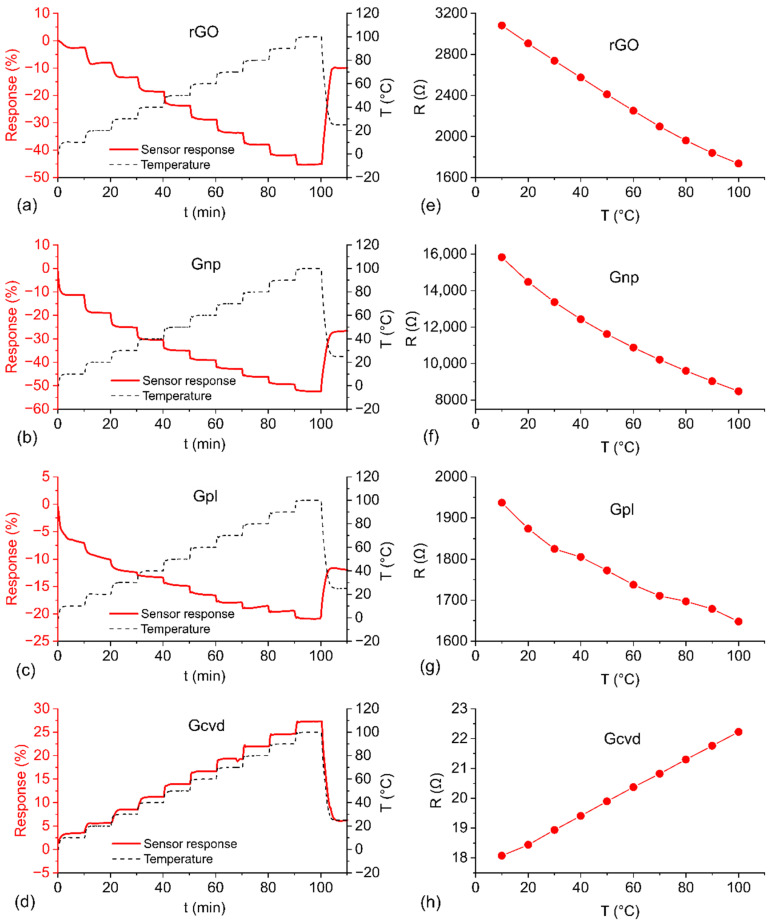
Summary of the time dependences of relative changes in resistance and temperature during climatic testing for (**a**) rGO, (**b**) Gnp, (**c**) Gpl, and (**d**) Gcvd and the temperature dependences of steady-state resistance at each step of testing for (**e**) rGO, (**f**) Gnp, (**g**) Gpl, and (**h**) Gcvd.

**Figure 5 nanomaterials-12-01594-f005:**
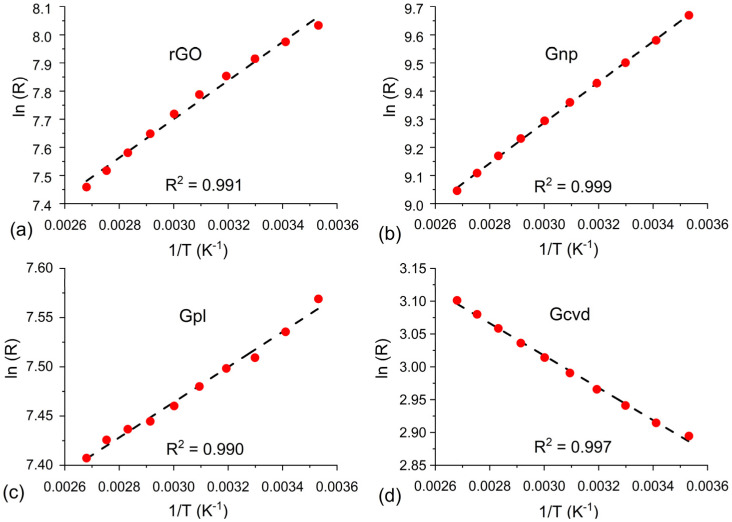
Logarithm of resistivity versus reciprocal temperature and confirmation of linearity, including the coefficient of determination, for (**a**) rGO, (**b**) Gnp, (**c)** Gpl, and (**d**) Gcvd.

**Figure 6 nanomaterials-12-01594-f006:**
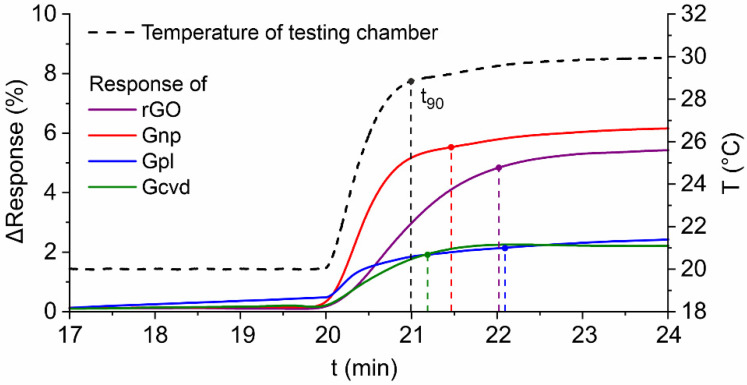
Sensors’ response time to temperature step change from 20 °C to 30 °C with the marked value *t_90_* for each sensor (and also temperature rise).

**Figure 7 nanomaterials-12-01594-f007:**
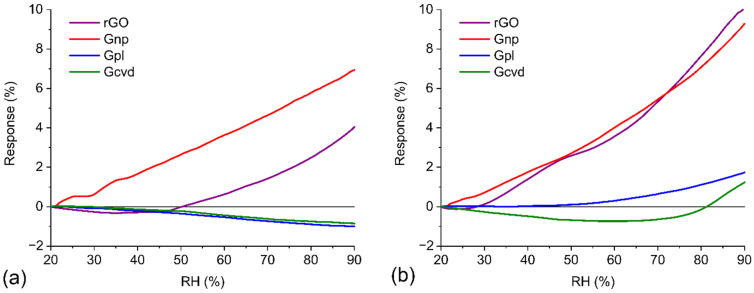
Humidity dependences of graphene-based materials (**a**) at 20 °C during the RH test and (**b**) at 50 °C during the RH test.

**Table 1 nanomaterials-12-01594-t001:** Atomic percentages of various elements in graphene-based materials.

Sample	Method	Carbon	Oxygen	Manganese	Aluminum
		At. Conc. (%)	At. Conc. (%)	At. Conc. (%)	At. Conc. (%)
rGO	EDS	47.58	42.80	5.05	4.56
Gnp	EDS	61.64	38.36	x	x
Gpl	EDS	76.03	23.97	x	x
Gcvd	XPS	99.14	0.86	x	x

**Table 2 nanomaterials-12-01594-t002:** Temperature and humidity parameters of graphene-based materials.

Samples	T Response 0–100 °C (%)	*TCR* (K^−1^)	*t_90_* (s)	Response to 20–90% RH at 20 °C (%)	Response to 20–90% RH at 50 °C (%)
rGO	−45.1	−4.86 × 10^−3^	121	4.1	10.1
Gnp	−52.0	−5.16 × 10^−3^	89	6.9	9.2
Gpl	−20.5	−1.68 × 10^−3^	125	−1.0	1.8
Gcvd	27.0	2.65 × 10^−3^	68	−0.9	|2.0|

**Table 3 nanomaterials-12-01594-t003:** Comparison of temperature dependences of graphene-based materials.

Materials	Temperature Range (K)	*TCR* (K^−1^)	Reference
Graphene oxide	303–373	−5.74 × 10^−3^	[27]
Conductive high strength metallurgical graphene	253–333	3.5 × 10^−4^ to 4.1 × 10^−4^	[31]
Semiconductive high strength metallurgical graphene	253–333	−1.7 × 10^−3^ to −4.0 × 10^−4^	[31]
Chemical vapor deposition graphene	253–333	1.5 × 10^−4^ to 4.0 × 10^−4^	[31]
Reduced graphene oxide monolayer	305–395	–9.5 × 10^−4^	[32]
Graphene ink	300–374	−1.77 × 10^−3^ ± 4.8 × 10^−4^	[33]
	173–300	−8 × 10^−4^ ± 5 × 10^−5^	[33]
Printed multilayer graphene film	6–360	−1.5 × 10^−4^	[34]
Graphene	303–373	−0.0105	[29]
TLG-coated fibers (trilayer graphene)	303–343	−0.0017	[35]
graphene nanoplatelets	213–333	−1.55 × 10^−3^ to −1.02 × 10^−3^	[36]
Graphene oxide	200–350	−0.024 to −0.04	[37]

## Data Availability

Not applicable.

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
