# Peer review of "Graphene-Based Temperature Sensors–Comparison of the Temperature and Humidity Dependences"

_nanomaterials, 2022, doi:10.3390/nano12091594_

Round 1

Reviewer 1 Report

  • Second author, OM, does not have affiliation, yet his address appears in institution 1
  • Suggestion: Rajan et al. show 2 graphene-based temperature sensors also with NTC and PTC for CVD graphene and NTC for graphene films with a more disordered structure (DOI:10.1021/acsami.0c08397) – this could be mentioned when you show similar behaviour for your 3 NTC compared with the PTC for CVD.
  • A photo or schematics of the devices would be very helpful, only details about a 100 micrometer width and spacing between the Au on Al interdigitated electrodes are given.
  • What is the temperature response of the IDEs on their own, i.e. before the graphene deposition? Has this been subtracted from the measured resistance for the sensors?
  • Not much detail about how the resistance was measured (contacts, method) – would be good to assess, for example, if the temperature response changes with applied voltage/current.
  • The oxygen atomic percentafe in rGO and Gnps make me question why these are called graphene in the first place. I’ve seen GO with similar oxygen to carbon ratios. Being so resistive (particularly Gnos), might be worth to try with GO directly to potentially make the process cheaper.
  • Authors mention “The main requirements are measurement accuracy, a good response time (e.g., low thermal inertia) and consistent performance under varying environmental conditions (mainly humidity).” (lines 63-65). A humidity study is done, the response is quantified, but there is no mention of response times. There seems to be a slight difference between when each temperature step is applied and the change in response from the sensor, but it is not clear by how much, or at least the order of magnitude of that response time.
  • How many cycles of temperature (e.g. 0-100 degrees Celsius, which is what is shown in Figure 2) can these sensors withstand? Same with humidity, do the sensors recover after being subject to high relative humidity values?
  • A simple encapsulation strategy could have been employed to prevent humidity from depositing at the graphene surface of the sensor. Please highlight why a humidity study is needed, then.
  • General review needed to eliminate typos. Minor editing examples: ensure superscripts in the relevant place, some are missing, e.g. 1 mm2 in line 132, Cu 2p3/2 and ½ in line 183. Line 300 “oxid” should be oxide.

Author Response

Dear Reviewer,

I am pleased to resubmit the revised version of nanomaterials-1681924 Graphene-based temperature sensors — comparison of the temperature and humidity dependences. We appreciate your constructive comments concerning our manuscript. We have studied your comments carefully and made corrections, and we hope that the revised manuscript meets your approval. We respond to your questions and comments in detail in the following text. Your comments from the review are in bold, normal font, and our responses are italicised.

Reviewer 1

1. Second author, OM, does not have affiliation, yet his address appears in institution 1.

Thank you for this remark. These errors were probably caused by our inattention.

2. Suggestion: Rajan et al. show 2 graphene-based temperature sensors also with NTC and PTC for CVD graphene and NTC for graphene films with a more disordered structure (DOI:10.1021/acsami.0c08397) – this could be mentioned when you show similar behaviour for your 3 NTC compared with the PTC for CVD.

Thank you for the recommendation, the article has been included in the overall comparison of temperature sensor properties (see Table x).

3. A photo or schematics of the devices would be very helpful, only details about a 100 micrometer width and spacing between the Au on Al interdigitated electrodes are given.

For better illustration, photographs of the entire sensor and a detail of the interdigital structure have been added to section 2.1 Sensor fabrication.

4. What is the temperature response of the IDEs on their own, i.e. before the graphene deposition? Has this been subtracted from the measured resistance for the sensors?

An interdigital structure without an active layer has no dependence on temperature (see attached graph). Furthermore, the active layers of temperature sensors have much lower resistance than the IDE structure itself (by 3 to 4 orders of magnitude). Thus, the temperature response of the sensor is determined only by the properties of the active graphene layer. (See attached graph)

5. Not much detail about how the resistance was measured (contacts, method) – would be good to assess, for example, if the temperature response changes with applied voltage/current.

A detailed description of the measurement method was added to the paragraph "2.3 temperature and humidity measurements": The sensors were contacted via spring test probes connected to the LRC meter using a 4-wire method. The test signal parameters were an AC voltage 0.25 V RMS and a frequency of 1000 Hz.

Furthermore, in the paragraph "3.2 Temperature dependences of graphene sensors", information about the sensor response at different bias voltage levels was added. For better understanding, a graph showing the sensor responses at different bias voltages has also been added.

6. The oxygen atomic percentafe in rGO and Gnps make me question why these are called graphene in the first place. I’ve seen GO with similar oxygen to carbon ratios. Being so resistive (particularly Gnos), might be worth to try with GO directly to potentially make the process cheaper.

This is an interesting point worth considering. Unfortunately, graphene oxide is not currently available for us to test and compare with other types of graphene. Furthermore, I assume that graphene oxide would have more resistance than the materials we use and we would probably have to change the electrode spacing, which could have a significant effect on the measured parameters. However, this is only an assumption.

7. Authors mention “The main requirements are measurement accuracy, a good response time (e.g., low thermal inertia) and consistent performance under varying environmental conditions (mainly humidity).” (lines 63-65). A humidity study is done, the response is quantified, but there is no mention of response times. There seems to be a slight difference between when each temperature step is applied and the change in response from the sensor, but it is not clear by how much, or at least the order of magnitude of that response time.

Thank you for your comment. In the section "3.2 Temperature dependence of graphene sensors", a paragraph describing the response time of the sensors has been added and a graph has also been added for full illustration. The graph shows the response time of each sensor to a step change in temperature.

8. How many cycles of temperature (e.g. 0-100 degrees Celsius, which is what is shown in Figure 2) can these sensors withstand? Same with humidity, do the sensors recover after being subject to high relative humidity values?

The repeatability of individual measurements was verified for both temperature and humidity dependence. The results can be seen in the attached graphs below. The repeatability for temperature cycling is more or less satisfactory for all sensors. In general, the sensors stabilize as the number of cycles increases. Two different trends can be seen from the moisture repeatability graph (figure 3). While a decreasing trend is seen for the rGO and Gnp materials when cycling between 20% RH and 80% RH, the Gpl and Gcvd materials show the opposite trend, i.e. an increasing trend. Moreover, the Gpl and Gcvd materials have a greater tendency to stabilize after a few measurement cycles.

Already after incorporating the revisions into the article, it is long with  quite a lot of figures. This is also the reason why we did not add these images, describing the repeatability of the sensors, to the final article. (See attached graph)

9. A simple encapsulation strategy could have been employed to prevent humidity from depositing at the graphene surface of the sensor. Please highlight why a humidity study is needed, then.

We have added a section of text to the introduction of the article where we tried to highlight why it is necessary to measure the humidity dependence of graphene-based temperature sensors.

You are definitely right that encapsulation is a simple way to limit the effect of humidity. On the other hand, it is another technological process in the sensor manufacturing and the materials used for encapsulation may not always be compatible with the active graphene layer. In addition, the materials used for encapsulation (various polymers) have low thermal conductivity, which can impair heat transfer to the active layer. This could result in a slower sensor response to temperature change. 

10. General review needed to eliminate typos. Minor editing examples: ensure superscripts in the relevant place, some are missing, e.g. 1 mm2 in line 132, Cu 2p3/2 and ½ in line 183. Line 300 “oxid” should be oxide.

Thank you for this remark. These errors were probably caused by our inattention. We have carefully reviewed the entire manuscript again and have corrected all these mistakes.

We spent many days applying revisions, as guided by your constructive comments. We appreciate the opportunity you have given us to improve our manuscript. We hope that the changes will meet with your approval.

Reviewer 2 Report

In this paper, the authors reported four different graphene temperature sensors and tested their temperature and humidity dependence. Energy-dis-persive X-ray diffraction was used to determine the elemental composition of the materials. Elemental analysis showed different oxygen contents in the structures of the materials. This paper is well written, but needs the following revisions before publication:

  1. What are the advantages of this job? The author needs to give a table and make a systematic comparison with the previous work.
  2. Why are the sensor performances of these four systems so different? The author needs to give a specific physical explanation.
  3. About “Graphene-based related devices”, some relevant literature authors need to mention, such as: RSC Adv., 2022, 12, 7821-7829; RSC Adv., 2017, 7, 25314-25324; Talanta, 2015, 134, 435-442; RSC Adv., 2019, 9, 41383-41391.

Author Response

Dear Reviewer,

I am pleased to resubmit the revised version of nanomaterials-1681924 Graphene-based temperature sensors — comparison of the temperature and humidity dependences. We appreciate your constructive comments concerning our manuscript. We have studied your comments carefully and made corrections, and we hope that the revised manuscript meets your approval. We respond to your questions and comments in detail in the following text. Your comments from the review are in bold, normal font, and our responses are italicised.

Reviewer 2

1. What are the advantages of this job? The author needs to give a table and make a systematic comparison with the previous work.

We apologies that this is not entirely clear from the article, we have tried to correct it and hope that it is now obvious.

The main objective of this paper is to compare different types of graphene each produced by a different method. To compare them with each other in terms of temperature and humidity dependence. Furthermore, to point out that the production method has a significant effect on both the structure of the material and especially on the above-mentioned dependencies. Last but not least, to make this comparison, which the literature dealing with graphene-based temperature sensors has been lacking.

Thank you for your comment. A table that compares our sensors with other sensors of this type in terms of their key features has been created and added to the article.

2. Why are the sensor performances of these four systems so different? The author needs to give a specific physical explanation.

Thank you for your comment. We realized that this explanation is not obvious, although it is given in section 3.2 describing the Arrhenius behavior of sensors and activation energies of materials. This explanation has been extended and improved to make the description of the different sensor response more understandable.

3. About “Graphene-based related devices”, some relevant literature authors need to mention, such as: RSC Adv., 2022, 12, 7821-7829; RSC Adv., 2017, 7, 25314-25324; Talanta, 2015, 134, 435-442; RSC Adv., 2019, 9, 41383-41391.

Thank you for the recommendations, some of this relevant literature, you recommended, has been incorporated into the introduction of the article.

We spent many days applying revisions, as guided by your constructive comments. We appreciate the opportunity you have given us to improve our manuscript. We hope that the changes will meet with your approval.

Round 2

Reviewer 2 Report

The article has been systematically modified and can be accepted.